# The Effect of FDI on Environmental Degradation in Romania: Testing the Pollution Haven Hypothesis

**Alexandru Chiriluș *** and Adrian Costea 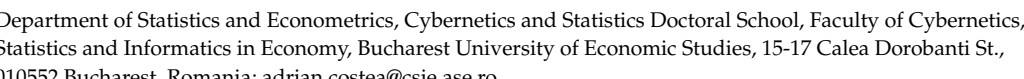

Department of Statistics and Econometrics, Cybernetics and Statistics Doctoral School, Faculty of Cybernetics, Statistics and Informatics in Economy, Bucharest University of Economic Studies, 15-17 Calea Dorobanti St., 010552 Bucharest, Romania; adrian.costea@csie.ase.ro
* Correspondence: chirilus.alexandru@gmail.com

**Abstract:** The study examines the relationship between $CO_2$ emissions, trade openness, GDP growth and foreign direct investment (FDI) in Romania. The research aims to provide empirical evidence for either the pollution haven hypothesis (PHH) or the pollution halo effect (PHE). The pollution haven hypothesis suggests that countries with weaker environmental regulations and lower environmental quality are more attractive to FDI, while the pollution halo effect posits that countries with high levels of environmental protection and quality can generate positive spillover effects for FDI. The findings suggest a significant relationship between $CO_2$ emissions, GDP growth and FDI inflows, with GDP growth having a greater effect on FDI than $CO_2$ emissions. GDP growth has a causal effect on $CO_2$ emissions, while $CO_2$ emissions have a causal effect on FDI. These findings have important policy implications, as they highlight the interplay between economic growth, environmental degradation, and foreign investment. Policies aimed at reducing emissions must be comprehensive and coordinated in order to achieve significant emissions reductions and strike a balance between economic growth and environmental protection.

**Keywords:** environmental degradation; foreign direct investment; pollution haven hypothesis; economic growth

## 1. Introduction

Foreign Direct Investment (FDI) is an investment made by a company or individual from one country into business interests located in another country. Generally, FDI takes place when an investor establishes foreign business operations or acquires foreign business assets, including establishing ownership or controlling interest in a foreign company. FDI is often used to take advantage of differentials in operating costs, access to natural resources, access to new markets, or access to financial and technological know-how [1].

FDI is often a major source of economic growth and development for countries that receive it [2]. It can bring new capital, technology, and expertise to the host country. It can also help to increase competition in the local market, which can lead to better products and services for consumers. FDI can also help to create new markets for goods and services produced in the host country, which can lead to increased exports and economic growth [3]. Additionally, FDI can help create new jobs in the host country, which can lead to increased incomes and improved living standards; however, such a relationship has been observed to be different from sector to sector [4]. Finally, FDI can help to improve infrastructure in the host country, which can lead to increased productivity and economic growth. It can also take many forms, including establishing a subsidiary or associate company in a foreign country, acquiring shares of an overseas company, or investing in a joint venture with a foreign partner. It can also involve the purchase of physical assets such as land, buildings, and equipment.

FDI is also an important source of capital for many countries, which can have a positive impact on economic growth and development. It can create jobs, increase productivity,

and stimulate economic activity. It can also bring new technologies and management practices to the host country [5]. If not appropriately managed, FDI can result in adverse consequences for the host country. These impacts may include:

1.  Loss of Control: FDI can lead to a loss of control over a country's economy. When foreign companies invest in a country, they often bring their own management and production techniques, which can lead to a loss of local control over the economy [6].
2.  Job Losses: FDI can lead to job losses in the host country. This is because foreign companies often bring their own workers with them, or they may outsource jobs to other countries with lower labor costs. In some cases, jobs created by FDI are more than offset by the jobs lost by domestic companies [6].
3.  Cultural Impact: FDI can have a negative impact on the culture of the host country. This is because foreign companies often bring their own values and practices, which can clash with the local culture [7]. Additionally, if it also leads to an influx of foreign workers, which can change the demographic makeup of certain areas and lead to cultural tension.
4.  Environmental Impact: FDI can have a negative impact on the environment of the host country [8]. When foreign companies invest in a country, they may establish industries that require large amounts of energy and natural resources. This increased demand for resources can lead to deforestation, water pollution, and other forms of environmental degradation.
5.  Political Influence: FDI can result in foreign companies exerting political influence in the host country due to their superior resources compared to local companies, giving them an unfair advantage in the political arena [9].

FDI can have both positive and negative impacts on the environment of the host country. On the positive side, FDI can bring in new technologies and capital that can help reduce pollution and improve environmental standards [5]. It can also create jobs and increase economic activity, which can lead to increased tax revenues that can be used to fund environmental protection initiatives. On the negative side, FDI has been observed to increase resource extraction, pollution, and environmental degradation. However, in some cases, the effect of industrialization on the environment was found to be statistically insignificant [10]. FDI can also lead to the displacement of local communities and destruction of natural habitats in low-income countries [11]. For example, the construction of mining or hydroelectric power plants can result in the clearance of forests or the alteration of natural waterways. Additionally, the expansion of industrial agriculture or urbanization can also contribute to habitat destruction. However, it is worth noting that the relationship between FDI and habitat destruction is complex and can also be influenced by a variety of factors, such as the specific industry or project, the location, and the regulatory environment. Additionally, FDI can lead to increased competition for resources [12], which can drive up prices and reduce access to resources for local communities.

FDI is commonly considered as a significant capital provider for numerous countries. Nonetheless, this source of funding is not free of consequences. We believe that FDI may have adverse effects on the environment, contributing to environmental degradation through various means, such as higher levels of $CO_2$ emissions, deforestation, and the destruction of natural habitats. One of the frequently observed negative impacts of FDI is environmental degradation, which is often caused by an increase in carbon emissions. This happens because foreign companies may not be subjected to the same environmental regulations as domestic companies, which gives them the opportunity to produce more pollutants than would be permitted otherwise [5]. This can lead to air and water pollution, which can have serious consequences for local ecosystems and human health. FDI can also lead to environmental degradation through deforestation. Numerous foreign companies clear extensive tracts of land for industrial or agricultural purposes, resulting in the obliteration of natural habitats and displacement of local wildlife [11]. This can have a devastating effect on local ecosystems and can also contribute to global climate change. Lastly, the destruction of natural habitats is another negative impact of FDI. Foreign companies

often construct factories or other infrastructure in areas that host endangered species or other delicate ecosystems. As a result, these habitats are destroyed, causing severe and long-lasting repercussions on biodiversity and the overall health of local ecosystems. This occurrence is particularly noticeable in low-income countries [11].

Overall, FDI has been observed having a significant impact on the environment, such as increased pollution, deforestation and the destruction of natural habitats. Ensuring accountability of foreign companies for their environmental impacts is crucial. It will aid in preventing further environmental degradation that could result from FDI. Therefore, it is essential for countries to take necessary measures to hold foreign companies responsible for their actions.

The pollution haven hypothesis (PHH) is an economic theory that suggests that companies will move their production to countries with weaker environmental regulations in order to reduce their costs. This is because the cost of complying with environmental regulations in the home country is higher than the cost of not complying in the host country [5]. This can lead to a race to the bottom, where countries compete to have the weakest environmental regulations in order to attract businesses. The hypothesis has been used to explain why some countries have higher levels of pollution than others. It suggests that companies are attracted to countries with weaker environmental regulations because they can save money by not having to comply with stricter regulations. This can lead to a situation where countries with weaker environmental regulations become pollution havens, while countries with stronger regulations become clean havens [13]. PHH has been criticized for its oversimplification of the issue, as it does not take into account other factors such as differences in economic development, infrastructure, and labor costs. Additionally, some argue that causality is more complex than a simple relationship between environmental regulation and FDI. Despite these criticisms, the hypothesis remains an important part of the discussion on global environmental regulation.

The phenomenon of FDI contributing to the reduction of environmental degradation is often referred to as the pollution halo effect (PHE). PHE suggests that multinational enterprises (MNEs) transfer their greener technologies or measures that reduce or control pollution, renewable energy-related and energy-conserving technologies, to their host countries. As a result, emissions are reduced in these countries, ultimately leading to a cleaner environment. Several studies that have found a positive relationship between FDI and the environment ("halo effect") that contrasted with the PHH, which argues that polluting industries often relocate production from countries with more strict environmental policies to countries with less strict ones, leading to a positive relationship between FDI inflow and $CO_2$ emissions. The impact of FDI on the environment can be heterogeneous across industries and countries, and the pollution haven and halo effects are often identified simultaneously [14]. A U-shaped relationship between environmental efficiency scores and regional GDP per capita levels has been observed in some studies, supporting PHE [15], which suggests that foreign enterprises have a positive effect on the environment. PHE has also been confirmed in some countries that had increasing FDI inflows over a long period of time with decreasing carbon emissions in both the short and long run [16].

Our research about the confirmation of PHH in Romania is important for a number of reasons. First and foremost, understanding the relationship between FDI and $CO_2$ emissions in Romania is crucial for policymakers as they work to create effective environmental policies. As stated, FDI can have in some cases a significant impact on the level of carbon emissions in a country, and our research helps to clarify the nature of this relationship for Romania. Additionally, by testing the PHH for Romania, our research makes a valuable contribution to the existing literature on this topic. The PHH suggests that countries with weaker environmental regulations will attract more foreign direct investment, as companies seek to take advantage of the lower costs associated with weaker environmental standards. Although previous research on this topic has yielded inconclusive results [8], our study aims to enhance our understanding of the connection between FDI and $CO_2$ emissions in Romania. Moreover, our findings could offer valuable insights for other nations that

encounter comparable difficulties in reconciling economic growth with environmental protection. As globalization continues to gain momentum, the correlation between FDI and emissions is becoming more critical for countries worldwide, and our investigation contributes to illuminating this crucial issue for Romania and other nations.

## 2. Literature Review

One study by Ging Lee [17] found that there is a short-term causal relationship between FDI inflows and pollution on output, measured by GDP per capita, in Malaysia. However, the relationship between output and FDI inflows is only validated in the long term. The study also suggests that the causality between FDI, pollution, and output may vary from country to country. The findings suggest that FDI inflows may act as a positive stimulus for Malaysia's economic growth, but they may not be a sustainable engine for growth in the long term. The study also suggests that GDP per capita has a positive effect on FDI inflows in the long run and that economic growth provides larger and growing markets for foreign firms. Additionally, the study finds that in the short term, FDI inflows play a significant role in environmental degradation.

Abdo [13] examined the relationship between FDI and environmental pollution in Arab countries. The results showed that a small increase in $CO_2$ emissions is influenced by FDI, supporting the idea that MNEs may move their investments or export waste to countries with less strict environmental regulations, confirming the PHH. The study also found that the direct effect of FDI is an increase in $CO_2$ emissions and environmental degradation. The authors also note that Arab countries are undergoing economic development and their economies are still in the early industrialization phase, with many manufacturing industries and a heavy dependence on nonrenewable energy sources such as coal and oil, which contributes to high levels of pollutant emissions. The study also discusses two different theoretical frameworks, the PHH and the PHE, and found that the evidence supports the former. It concludes that FDI has a negative effect on environmental pollution in Arab countries, and that the weak environmental laws and extractive industries in these countries contribute to this relationship. Their research also stresses the need for stricter environmental regulations and policies in Arab countries to reduce the negative effects of FDI on pollution.

Research carried out by Apergis [18] on the relationship between FDI flows and $CO_2$ in developing countries, specifically focusing on the BRICS countries (Brazil, Russia, India, China, and South Africa), referenced the two hypotheses. The findings of the study confirm that FDI from Denmark and the UK to BRICS countries leads to an increase in carbon emissions, thereby supporting the PHH. Conversely, FDI from France, Germany, and Italy leads to a reduction in carbon emissions, indicating support for the PHE. The study acknowledges that while FDI brings knowledge spillovers, improved institutional quality, and economic growth, it also results in increased environmental degradation. (Knowledge spillovers refer to the unintentional transfer or sharing of knowledge, expertise, or ideas from one entity to another. In the context of FDI, knowledge spillovers can occur when MNEs bring new technologies, management practices, or R&D to the host country. This can benefit the local economy by promoting innovation, productivity, and competitiveness. Additionally, employees of the MNE may acquire new skills and knowledge, which they can later use to start their own businesses or contribute to the local economy. Thus, knowledge spillovers resulting from FDI can have positive effects on the host country's economic growth and development.) The study proposes that future research could investigate whether FDI flows to specific sectors lead to environmental degradation in the host country, as recent research by Casino [19] suggests that the PHH only holds true in primary and manufacturing sectors rather than when aggregate FDI inflows are considered.

Tang [20] found that countries with stricter environmental regulations than the United States tend to have a stronger effect on FDI and that export-oriented FDI is more sensitive to local environmental regulations than local market-oriented FDI. (Export-oriented FDI is when MNEs invest in a host country to produce goods or services for export to other

markets. The primary goal of export-oriented FDI is to take advantage of the lower costs of production in the host country, such as cheaper labor, land, or raw materials, and to export the produced goods or services back to the home country or other markets where they can be sold at a higher price. Local market-oriented FDI is when MNEs invest in a host country primarily to serve the local market, which means to sell goods or services to local customers. In this type of FDI, the foreign firm is seeking to expand its market share by accessing the host country's market, which can offer growth opportunities for the corporation.) They also suggest that pollution-intensive industries are more likely to be attracted to countries with lax regulations and that the lack of capital available in these countries can generate an opposite force to the PHH. The author also mentions that FDI is affected by factors such as market potential, local policies, and the environmental policies of neighboring countries. They cite previous studies that have shown that FDI can facilitate economic growth through knowledge and technological spillovers. Overall, the author suggests that stricter environmental regulations in adjacent countries can make a host country more favorable for FDI.

Bulus [21] discusses the relationship between FDI and $CO_2$ emissions in Korea over the period of 1970–2018. The study investigates the dynamic interaction among variables and the validity of two hypotheses: the Porter hypothesis and the PHH. (The Porter hypothesis suggests that stricter environmental protections can have a positive effect on firms' performance by encouraging innovation.) The study finds that increased FDI, per capita GDP, energy use, and imports have led to increased per capita $CO_2$ emissions in Korea, while government expenditures, renewable energy, and exports have led to decreased per capita $CO_2$ emissions. The study finds that government expenditures have a negative effect on $CO_2$ emissions over the long term, and renewable energy and exports have negative coefficients, meaning they are environmentally beneficial. Imports have a positive long-term coefficient, meaning they increase per capita $CO_2$ emissions. Overall, the study suggests that FDI, GDP, energy use, and imports lead to increased per capita $CO_2$ emissions in Korea, while government expenditures, renewable energy, and exports lead to decreased per capita $CO_2$ emissions, and it could not provide robust evidence for the PHH in Korea.

Waldkirch [22] examined the extent to which pollution intensity of production helps explain FDI in Mexico. The study found that there is a positive correlation between FDI and pollution that is statistically and economically significant in the case of highly regulated sulfur dioxide emissions. Industries for which the estimated relationship between FDI and pollution is positive receive up to 30% of total FDI and 30% of manufacturing output. The study also considered the endowment of factors such as skilled labor and capital, which largely determined where industry is located and which goods a country will export. The research found that there is indeed evidence of a pollution haven effect but only for sulfur dioxide emissions and only for industries with large firms. However, the study also points out that inducing one or a small group of countries to tighten regulation is likely to simply shift the problem of pollution-intensive production to other favorable developing-country hosts over time.

Solarin [23] investigated the correlation between foreign direct investment and $CO_2$ emissions in Ghana for the period of 1980–2012. The study employs a plethora of variables as determinants of $CO_2$ emissions, including GDP, GDP square, energy consumption, renewable energy consumption, fossil fuel energy consumption, FDI, institutional quality, urbanization, and trade openness. The results demonstrate that GDP, FDI, urban population, financial development, and international trade have a positive impact on $CO_2$ emissions, while institutional quality decreases emissions in Ghana. The study also aimed to test for the presence of an Environmental Kuznets Curve (EKC) (see Figure 1). (The EKC postulates that as a country's income increases, its environmental degradation first increases and then decreases.) The study ascertains that energy consumption is positively associated with $CO_2$ emissions and that reducing energy consumption will decrease carbon emissions. It also notes that the transportation sector and the energy consumption of houses

and buildings contribute to emissions, and that fossil fuels are responsible for the largest quantity of emissions while renewable energy contributes very minimal quantity. The study concludes that Ghana has developed a comparative advantage in pollution-intensive industries and become one of the "havens" for the world's polluting industries due to its weaker environmental policy. The study suggests that Ghana should augment its environmental policy to reduce the negative impacts of foreign direct investment on emissions.

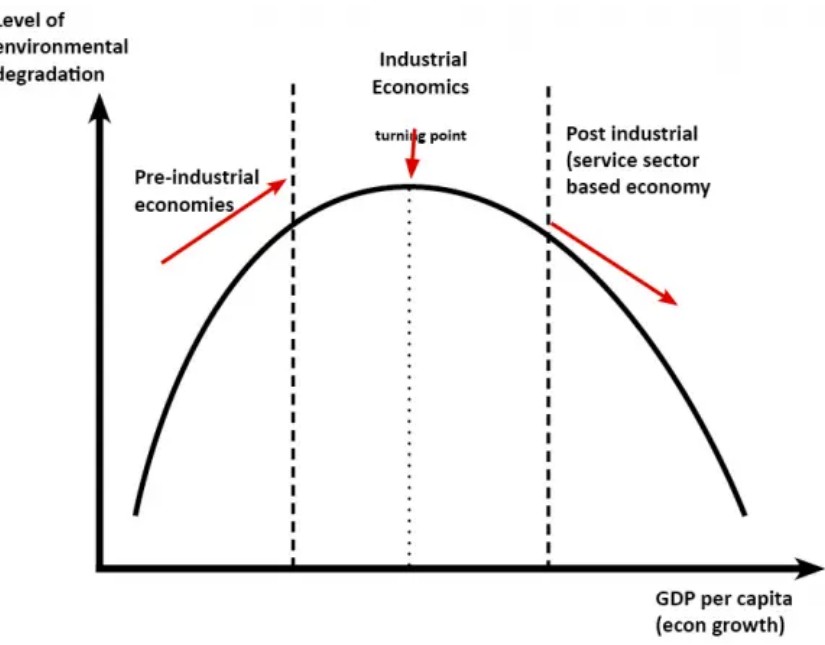

**Figure 1.** Graphical representation of the Environmental Kuznets Curve (EKC); Source: economichelp.com, accessed on 20 November 2022.

Khan [24] investigates the causal linkage between environmental pollution by $CO_2$ emissions and FDI along with other variables such as economic growth by real per capita income and trade openness, using balanced annual data from 17 countries in Asia for the period from 1980 to 2014. The study finds that inward FDI has a significantly positive impact on environmental pollution, supporting the PHH. It also finds that economic growth and trade openness are pivotal determinants of FDI. The study argues that economic policy reforms are required to channel foreign capital inflows to a more environmentally healthy direction. The study also notes that while economic growth has lifted many individuals out of poverty and improved income levels, it often comes at the cost of environmental degradation and poorer public health. In addition, Singhania [25] explored the relationship between FDI, institutional factors, financial development, and sustainability by revisiting the pollution haven (or halo) hypotheses. The study uses data from the World Development Indicators (WDI) database over the period of 1990–2016, covering 21 developed and developing countries with high carbon emissions. Their findings are that FDI has a significant positive impact on environmental degradation, providing evidence of PHH, particularly in developing countries. The study notes that MNEs tend to exploit the benefits of internationalizing firm-level advantages such as technical skills, advertising, and brand names, but this can lead to environmental degradation. The study also notes that developing countries with lower infrastructural standards may compromise on the use of dirtier technology in order to attract FDI, leading to the PHH. In conclusion, both studies highlights the need to consider financial development, institutional frameworks, and foreign capital inflows in order to achieve sustainable development. Additionally, it is emphasized that weak or ineffective environmental laws and regulations in a recipient country could attract incoming FDI by profit-driven companies willing to avoid pricey/expensive regulatory agreements

in the country of origin, and that economic policy reforms are required to channel foreign capital inflows to a more environmentally healthy direction.

A sizable amount of literature has developed which analyzes trade and investment flows to find support for the PHH. The literature has mainly been focused on the effect of FDI within a country on $CO_2$ emissions. However, Manderson [26] claims that very little robust evidence has been found in favor of the PHH. Their study contributes to the literature by looking at FDI outflows and focusing on the effect of differential regulations between countries. In particular, the study accounts for heterogeneous firm behavior, which has been shown to be important in explaining the structure of international trade and investment but has been largely overlooked by studies of pollution havens. (Heterogeneous firm behavior refers to the idea that firms within an industry or sector can have different characteristics, such as different levels of environmental compliance costs, and that these differences can affect their behavior and decisions related to trade and investment.) The study finds that modeling FDI in this way does not uncover evidence of a pollution haven effect. UK firms which find it costly to comply with environmental regulation are not more likely to establish foreign subsidiaries than those firms with low environmental compliance costs. The study also finds that high environmental cost MNEs do not have a greater propensity to locate subsidiaries in host countries with lax environmental policy than low-cost MNEs, controlling for a range of other interaction terms between environmental costs and host country characteristics. The study concludes that the lack of widespread evidence of pollution havens does not arise because they are simply being aggregated away by ignoring within-sector differences in firm characteristics, and it highlights important differences in the FDI behavior of high and low environmental cost firms.

Another study by Eskeland [27] discovers no discernible correlation between the levels of FDI in some nations and the costs of reducing pollution in developed countries. Their research also examined whether MNEs within industries tend to exhibit higher or lower levels of pollution in comparison to their domestic counterparts. Utilizing energy consumption and the utilization of "dirty fuels" as proxies for pollution intensity, the study finds that foreign plants are significantly more energy efficient and rely on cleaner forms of energy. Furthermore, the study finds that MNEs are less polluting than their domestic peers in developing countries. The conclusion of these findings imply that policy makers should prioritize controlling pollution itself rather than investment or specific investors. Additionally, the study notes that the lack of evidence for the PHH does not imply that such a phenomenon cannot exist or that pollution in developing countries should not be a concern.

Rezza [28] compared the amount of investment received by subsidiaries of MNEs located in countries with stricter environmental regulations to those located in countries with more lenient regulations. The study finds that the efficiency-seeking subsidiaries located in countries with stricter environmental regulations receive less investment from their parent companies in terms of equity capital, capital stock, and assets. The study contributes to the literature on the determinants of FDI and pollution havens in two ways. First, it demonstrates that parent companies invest less in subsidiaries with vertical motives in countries with stricter environmental regulations. (Vertical motives refer to a type of motive behind FDI in which MNEs invest in a host country to take advantage of lower costs or to access certain resources that are not available or are more expensive in their home country.) Second, it shows that the production of subsidiaries that is exported back to the parent company decreases with the strictness of the country's environmental regulations and their enforcement. While the effect of regulations on multinationals' activities is economically significant, it remains to be seen whether the incoming FDI that countries obtain because of lenient environmental regulations will also be economically significant for the country. For example, Aminu [29] finds that "dirty" FDI outflow is positively correlated with environmental policy in the eleven OECD countries, but FDI inflow is not significant in explaining the level of pollution and energy use in the fourteen non-OECD countries.

This suggests that MNEs are more likely to relocate production to countries with weaker environmental policies, but that this does not necessarily lead to increased pollution

or energy use in those countries. The study highlights that many factors can influence MNEs' relocation decisions and suggests that consistent environmental regulation, rather than lax policy, could be more effective in addressing the PHH.

Soilita [30] used data from the WDI (World Development Indicators) database over the period of 1990–2016, covering 21 developed and developing countries with high carbon emissions. The study finds that FDI has a significant positive impact on environmental degradation, providing evidence of the PHH, particularly in developing countries. The study also finds that both pollution haven and factor endowments hypotheses act simultaneously, with opposing effects in different types of countries, and it highlights the need to consider financial development, institutional frameworks, and foreign capital inflows in order to achieve sustainable development.

A summary of the key points:

- The pollution haven hypothesis (PHH) proposes that companies will relocate production to nations with looser environmental rules to cut costs.
- The pollution halo effect (PHE) refers to the reduction of environmental degradation through FDI flows.
- Lack of Sovereignty: Foreign Direct Investment (FDI) may result in a diminished sovereignty over a nation's economy.
- An understanding of the relationship between foreign direct investment (FDI) and carbon emissions ($CO_2$) in Romania is important for policymakers to create effective environmental policies.

### 3. Data, Model and Methodology

#### 3.1. Data Description

The data used in this study are based on annual data collected over the period of 2003 to 2021. However, in the analysis of FDI groups, the data collected span over a period of 15 years, starting from 2007 and extending until 2021. This time frame offers information that provides an extensive understanding of the evolution and fluctuations of the subject matter. With these data, we can analyze the causality that has developed over the years, providing a comprehensive picture of the evolution of $CO_2$ emission for Romania. This information is valuable in gaining insights into the underlying factors that influence the level of carbon emissions in the host country and can be utilized to make informed decisions and formulate strategies for the future. The annual data provide a snapshot of the situation at a given time and allow for an examination of changes and developments over the years. The sample size is $n = 19$ ($n = 15$ in the case of FDI groups). The study uses the following variables: FDI inflows by groups ($FDI_h$), environmental pollution measured in carbon emissions by type ($CO_{2,i}$), economic growth as GDP per capita ($GDP_t$) and trade openness ($TO_t$). (The formula for Trade Openess ($TO_t$) is as follows: $\frac{X+M}{GDP}$, where X = Exports; M = Imports). Data on most of the selected variables have been obtained from The National Bank of Romania (https://bnr.ro/Interactive-database-1107.aspx, accessed on 21 November 2022). For $CO_2$ emissions, data have been gathered from (https://ourworldindata.org/co2/country/romania, accessed on 21 November 2022).

#### 3.2. Unit Root Test

To assess the integrating properties of the data before conducting statistical analysis, three widely used panel unit root tests were employed in this study.

These tests include:

- Augmented Dickey–Fuller (ADF);
- Phillips–Perron (PP);
- Kwiatkowski–Phillips–Schmidt–Shin (KPSS).

Taking the first difference of a non-stationary time series involves computing the difference between each observation and the previous one. This is completed to remove any trend or seasonality present in the data, which may cause the series to be non-stationary. Non-stationary time series have statistical properties, such as the mean and variance, that

change over time and are not constant. However, in our case, after taking the first difference of these variables, they became stationary. This is because taking the difference between each observation and the previous one removes any trends or seasonality that may be present in the data. As a result, the statistical properties of the time series became constant over time, making them stationary.

### 3.3. Wavelet Analysis

Wavelets (see Figure 2) can decompose a time series into more elementary functions, allowing for useful information to be retrieved from the signals [31]. Wavelet coherence analysis is a method used to analyze the interdependence between two time series signals in the frequency domain. It is an extension of wavelet analysis and provides information about the degree and direction of the relationship between the two signals. The coherence function is calculated using the cross-spectral density of the signals and the power spectral density of each individual signal. The result of the coherence function is a measure of the coherence between the two signals, with values ranging from 0 to 1, where 1 (in red) indicates a strong and consistent relationship between the signals.

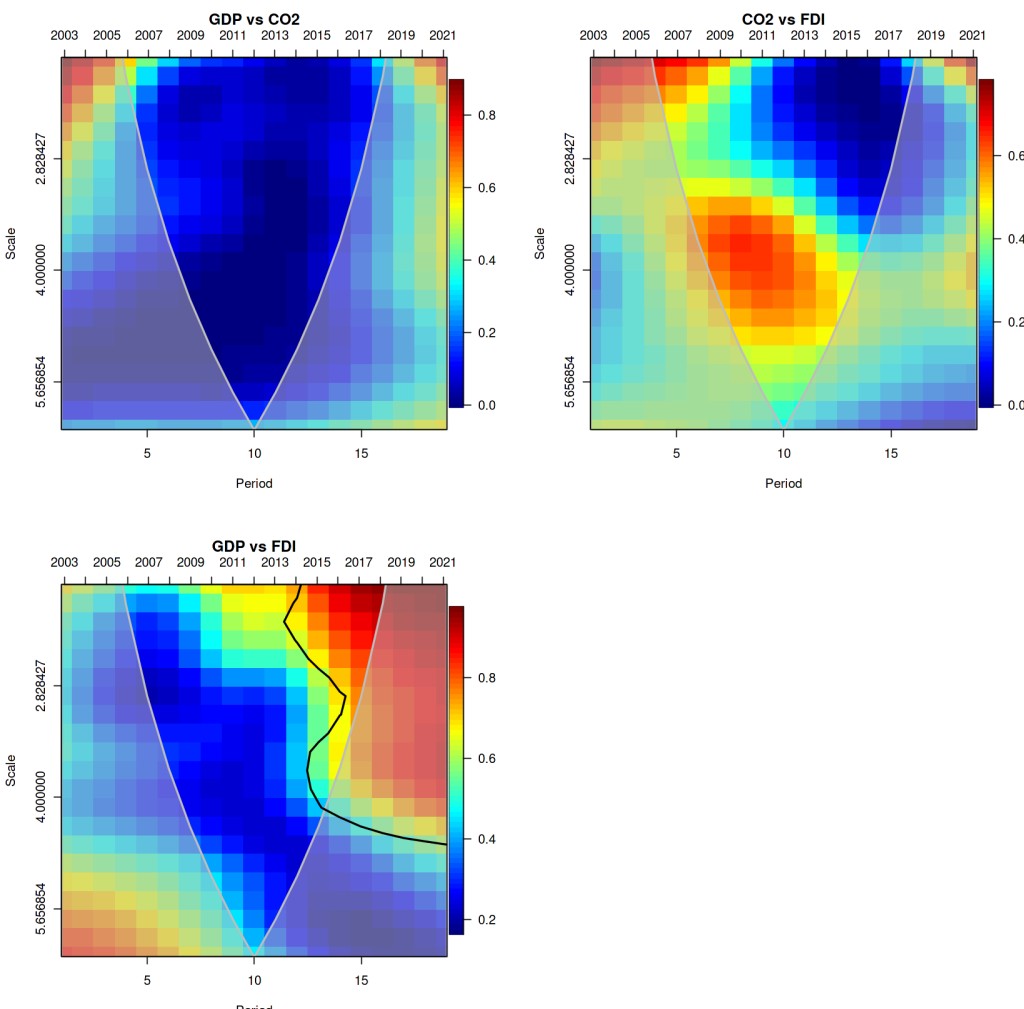

**Figure 2.** Wavelet coherence analysis visual plot; Source: author's calculation.

### 3.4. Linear Model

The present study aims to investigate the causal relationship between FDI inflow, environmental pollution ($CO_2$ emissions), economic growth, and trade openness through a

rigorous empirical analysis. Two sets were created, one for each type of FDI and another for each type of carbon dioxide ($CO_2$) emissions (see Equation (1)).

$$CO_{2,i} = \{Oil, Gas, \dots\}; \; i = \overline{1,6}$$
$$FDI_h = \{FDI_{extr}, FDI_{prel}, \dots\}; \; h = \overline{1,11} \tag{1}$$

Correlation Matrix

A simple linear regression model was then applied to both sets in order to establish their correlation matrix (see Equation (2)). The aim of this approach was to investigate whether there was a correlation between FDI and $CO_2$ emissions by analyzing their linear relationship. The correlation matrix serves as a visual representation of the strength and direction of the relationship between the two variables, providing valuable insights into their interdependence.

$$CO_{2,i,t} = \gamma_0 + \gamma_1 FDI_{h,t} + \varepsilon_t \tag{2}$$

The research also utilized another mathematical framework to express the relationship between the selected variables, which is represented by the following functional form (see Equation (3)) [24]. This functional form serves as a foundation for the study's empirical investigation, which will provide valuable insights into the complex and interrelated dynamics between FDI, environmental pollution through carbon emissions, economic growth, and trade openness. The results of this study have the potential to inform policy decisions and contribute to the advancement of the academic literature in this field.

$$CO_2 = f(FDI, GDP, TO)$$
$$FDI = f(CO_2, GDP, TO) \tag{3}$$

The current model draws inspiration from various similar studies that have been conducted in the past. However, it aims to present a unique perspective and build upon the established theories, incorporating new insights and considerations. This creates a nuanced and comprehensive approach to the subject matter, contributing to a deeper understanding of the interplay between different variables and their impact on $CO_2$ emissions and foreign direct investment inflows.

Equation (3) can be rewritten in the following linear form:

$$CO_{2,t} = \beta_0 + \beta_1 FDI_t + \beta_2 GDP_t + \beta_3 TO_t + \varepsilon_t \tag{4}$$

$$FDI_t = \theta_0 + \theta_1 CO_{2,t} + \theta_2 GDP_t + \theta_3 TO_t + \varepsilon_t \tag{5}$$

In Equation (2), the $i$ ranges from 1 to 6 and the $h$ ranges from 1 to 11. Similarly, the index $t$ ranges from 1 to 15, representing the time period under consideration. The coefficients $\beta_i$ and $\theta_i$ in Equations (4) and (5) represent the regression coefficients. In Equations (6) and (7), the parameters $k$ and $p$ correspond to the optimal lagged values for the respective term. The term $\varepsilon_t$ represents the residual error in each of the model equations.

Previous research has established a positive correlation between FDI inflows and variables such as GDP per capita, trade openness and environmental degredation measured by $CO_2$ emissions. This is due to weak regulations and a lack of enforcement, leading to greater investment in countries with higher levels of pollution [20]. Studies also suggest a positive relationship between $CO_2$ emissions, FDI inflows, economic growth, and trade openness [24].

### 3.5. Conducting Granger-Causality Analysis

Granger causality is a concept that assesses the causal relationship between two time-series variables. It is based on the idea that the past values of a variable can help to predict its future values, but the reverse is not necessarily true. In other words, if variable $X$ Granger causes variable $Y$, then the information contained in the past values of $X$ can be used to improve the forecast of Y but not vice versa [32]. This causal relationship is determined

by performing a regression analysis and testing the significance of the coefficients of the lagged values of $X$ in the regression equation for $Y$.

The Granger causality multivariate regression model of this study can symbolically be expressed as follows:

$$CO_{2,t} = \alpha_0 + \sum_{i=1}^{k} \beta_i CO_{2,t-i} + \sum_{i=1}^{k} \theta_i FDI_{t-i}$$
$$+ \sum_{i=1}^{k} \gamma_i GDP_{t-i} + \sum_{i=1}^{k} \theta_i TO_{t-i} + \varepsilon_{1,t}$$
(6)

$$FDI_t = \omega_0 + \sum_{i=1}^{p} \epsilon_i FDI_{t-i} + \sum_{i=1}^{p} \zeta_i CO_{2,t-i}$$
$$+ \sum_{i=1}^{p} \delta_i GDP_{t-i} + \sum_{i=1}^{p} \psi_i TO_{t-i} + \varepsilon_{2,t}$$
(7)

## 4. Results

### 4.1. Unit Root Test

In Table 1, the results of tests for unit roots are presented, which are used to determine whether a time series is stationary or non-stationary. The majority of the selected variables, including FDI inflows, $CO_2$ emissions, economic growth and trade openess, were found to be non-stationary at the level. This means that these variables have statistical properties that vary over time and are not constant.

It should be noted that one variable, $FDI_{finan}$, did not become stationary after taking its first difference. This suggests that this variable may have a more complex statistical structure that cannot be captured by a simple differencing operation.

**Table 1.** Panel unit root analysis.

| Variable | Description | At Level | | | First Difference | | |
| --- | --- | --- | --- | --- | --- | --- | --- |
| | | ADF | PP | KPSS | ADF | PP | KPSS |
| $FDI_{extr}$ | Mining | −1.7811 | −10.0140 | 0.0800 | −2.4810 | −13.3589 | 0.1089 |
| $FDI_{prel}$ | Manufacturing | −0.5000 | −14.5779 | 0.1784 | −2.1671 | −21.7338 | 0.2379 |
| $FDI_{ener}$ | Electricity | −2.2953 | −10.9503 | 0.4340 | −4.2775 | −14.5195 | 0.1448 |
| $FDI_{agri}$ | Agriculture | −1.0948 | −11.2470 | 0.1215 | −3.5453 | −15.9880 | 0.1468 |
| $FDI_{cons}$ | Construction | −4.7551 | −11.3845 | 0.1344 | −5.1057 | −18.9254 | 0.0959 |
| $FDI_{tech}$ | Technology | −1.4941 | −13.7935 | 0.1626 | −2.3667 | −20.2430 | 0.1643 |
| $FDI_{comr}$ | Trade | −0.1617 | −6.9277 | 0.3293 | −3.6802 | −17.0656 | 0.3949 |
| $FDI_{hote}$ | Accomodation | −3.9576 | −11.8331 | 0.1372 | −4.5071 | −12.4891 | 0.1147 |
| $FDI_{trans}$ | Transportation | −0.7678 | −17.2211 | 0.0963 | −4.1161 | −17.6494 | 0.1218 |
| $FDI_{finan}$ | Finance | 0.3216 | −6.5362 | 0.2005 | −1.6790 | −7.4031 | 0.5088 |
| $FDI_{activ}$ | Admin. activities | −3.1267 | −7.2177 | 0.3889 | −3.9758 | −17.8704 | 0.2556 |
| $FDI$ | Total | −0.2100 | −11.4116 | 0.1102 | −2.1646 | −24.1878 | 0.1244 |
| Oil | $CO_2$ emissions | 0.1266 | −3.3298 | 0.1839 | −2.7155 | −15.4897 | 0.3652 |
| Gas | $CO_2$ emissions | −1.2336 | −3.8457 | 0.6472 | −3.4127 | −19.2604 | 0.2366 |
| Flaring | $CO_2$ emissions | −0.5996 | −9.1304 | 0.1677 | −3.5403 | −16.1959 | 0.0707 |
| Coal | $CO_2$ emissions | −1.2322 | −0.4452 | 0.6758 | −4.3051 | −12.7722 | 0.1464 |
| Cement | $CO_2$ emissions | 0.1596 | −7.6874 | 0.1657 | −3.6095 | −11.9212 | 0.1041 |
| Others | $CO_2$ emissions | −0.3931 | −10.1002 | 0.2044 | −4.0385 | −14.5523 | 0.0702 |
| $CO_2$ | Total | −0.9418 | −2.3531 | 0.6471 | −4.6660 | −12.8169 | 0.1183 |
| $GDP$ | Per capita PPP | 2.8826 | 0.6049 | 0.7212 | −0.5505 | −15.3711 | 0.2136 |
| $TO$ | Trade openess | 0.8636 | −1.1867 | 0.6000 | −2.4415 | −16.3495 | 0.1398 |

Source: Author's calculation.

### 4.2. Wavelet Analysis

The wavelet graph (see Figure 2) reveals a degree of intensity between $CO_2$ and FDI during the period from 2003 to 2013. The graph shows that there is a moderate correlation between the two variables, with high intensity levels present throughout most of the time period. The intensity levels decrease slightly in 2008 and 2009 before rising again toward the end of the period. Similarly, the wavelet graph displays the degree of intensity between GDP and FDI between 2015 and 2021. The graph indicates a moderate to high correlation between these two variables with non-varying levels of intensity throughout the period. The intensity levels increase in 2015 and 2017 before increasing even more slightly toward the end of the period. Overall, the wavelet graph provides a visual representation of the relationship between these economic indicators and highlights the degree of intensity (variables moved in the same direction) and trends that have occurred over time.

### 4.3. Linear Model

Since correlation analysis shows how strongly two variables are linked and to what extent they tend to change together, the correlation matrix presented in (Table 2) highlights the relationship between each FDI group and each $CO_2$ emission type. The *t*-test matrix in (Table 3) shows the statistical significance (A t-distribution table of critical values was used for the interpretation of statistical significance for each correlation coefficient) for each correlation coefficient. After conducting a thorough analysis, with the exception of $FDI_{prel}$ and Oil; $FDI_{cons}$ and Cement; it can be concluded that there are no other statistical significant correlations between the other FDI groups and $CO_2$ emissions. This means that the combination of these variables does not produce any meaningful or noteworthy results. This result suggests that most FDI groups are not linked to different types of $CO_2$ emissions. Ultimately, the findings from the correlation matrix reinforce the need for a more comprehensive and multi-faceted approach to understanding the relationship between FDI and carbon emissions.

**Table 2.** Correlation matrix.

| Variable | Oil | Gas | Flaring | Coal | Cement | Others |
|---|---|---|---|---|---|---|
| $FDI_{extr}$ | 0.0085 | −0.1704 | 0.1432 | −0.1655 | −0.3270 | 0.0359 |
| $FDI_{prel}$ | 0.5245 | 0.4082 | 0.1561 | 0.2874 | 0.2085 | 0.2135 |
| $FDI_{ener}$ | 0.3446 | 0.2876 | 0.1524 | 0.1843 | 0.1715 | −0.2505 |
| $FDI_{agri}$ | 0.1963 | −0.0982 | 0.0987 | 0.3624 | 0.1965 | −0.1623 |
| $FDI_{cons}$ | 0.3984 | 0.4586 | −0.0865 | 0.4903 | 0.5254 | −0.2441 |
| $FDI_{tech}$ | 0.4728 | −0.0277 | 0.2034 | 0.4418 | 0.3082 | −0.0145 |
| $FDI_{comr}$ | 0.3715 | 0.0225 | 0.3707 | 0.3313 | 0.2760 | 0.1861 |
| $FDI_{hote}$ | −0.0015 | −0.3202 | −0.4337 | −0.0651 | −0.1358 | 0.1098 |
| $FDI_{trans}$ | −0.1660 | −0.6454 | −0.1792 | −0.1382 | −0.0106 | −0.2525 |
| $FDI_{activ}$ | 0.2711 | 0.3438 | 0.2950 | 0.4139 | 0.3905 | 0.1713 |

Source: Author's calculation.

**Table 3.** Test for statistical significance.

| Variable | Oil | Gas | Flaring | Coal | Cement | Others |
|---|---|---|---|---|---|---|
| $FDI_{extr}$ | 0.03 | −0.62 | 0.52 | −0.61 | −1.25 | 0.13 |
| $FDI_{prel}$ | 2.22 | 1.61 | 0.57 | 1.08 | 0.77 | 0.79 |
| $FDI_{ener}$ | 1.32 | 1.08 | 0.56 | 0.68 | 0.63 | −0.93 |
| $FDI_{agri}$ | 0.72 | −0.36 | 0.36 | 1.40 | 0.72 | −0.59 |
| $FDI_{cons}$ | 1.57 | 1.86 | −0.31 | 2.03 | 2.23 | −0.91 |
| $FDI_{tech}$ | 1.93 | −0.10 | 0.75 | 1.78 | 1.17 | −0.05 |
| $FDI_{comr}$ | 1.44 | 0.08 | 1.44 | 1.27 | 1.04 | 0.68 |
| $FDI_{hote}$ | −0.01 | −1.22 | −1.74 | −0.24 | −0.49 | 0.40 |
| $FDI_{trans}$ | −0.61 | −3.05 | 0.66 | −0.50 | −0.04 | −0.94 |
| $FDI_{activ}$ | 1.02 | 1.32 | 1.11 | 1.64 | 1.53 | 0.63 |

Source: Author's calculation.

The results shown in Table 4 suggest that there is a relationship between $CO_2$ emissions and FDI. The *p*-value of 0.0701 indicates that there is a 7.01% chance that the relationship between the variables in the model is due to random chance, which is slightly above the commonly accepted significance level of 0.05. Therefore, there may be some evidence to suggest that the relationship between $CO_2$ emissions, FDI, GDP and TO is statistically significant. In the case of the second regression model, this suggests that there is a relationship between the level of FDI and the variables of $CO_2$ and GDP. The low *p*-value of 0.0061 indicates that there is a very small chance (0.61%) that the relationship between the variables in the model is due to random chance; therefore, there is strong evidence to suggest that there is a relationship between FDI, $CO_2$ emissions, and GDP. The directionality of this relationship cannot be determined from the given information. However, the model provides a useful framework for exploring the relationship between these economic indicators and can inform decision making in the areas of policy and investment.

**Table 4.** Linear regression model.

| $Y_t$ | $X_t$ | *p* Value |
|---|---|---|
| $CO_{2,t}$ | $1166FDI_t^* + 854GDP_t - 8393631TO_t - 2902603$ | 0.0701 |
| $FDI_t$ | $0.0001CO_{2,t}^* + 1.26GDP_t^* + 9156TO_t - 1537$ | 0.0061 |

(*) *p*-value < 0.1; Source: Author's calculation.

*4.4. Granger Causality*

**Hypothesis 0.** *Does NOT Granger-cause.*

The results of the causality model (see Table 5) provides valuable insights into the relationship between FDI inflows, $CO_2$ emissions, GDP and trade openness. Our findings indicate that FDI inflows do not Granger-cause $CO_2$ emissions. However, the same model shows that GDP does Granger-cause $CO_2$ emissions in Romania, confirming the EKC (see Figure 1). This suggests that economic growth and development, as measured by GDP, do have an influence on carbon emissions. The result that trade openness does not cause $CO_2$ emissions is also noteworthy, as it indicates that trade policies and practices may not be as influential in determining carbon emissions levels as previously thought. Our study also found that GDP and trade openness do not cause FDI inflows. This suggests that other factors, such as political stability, access to resources and infrastructure, play a more significant role in attracting foreign investment. However, the results also show that $CO_2$ emissions do cause FDI inflows. This finding is particularly interesting as it suggests that companies and investors may consider environmental factors when making investment decisions, confirming PHH in Romania. The results of this model offer a detailed examination of the relationship between FDI, $CO_2$ emissions, GDP, and trade openness. The findings suggest that there is no straightforward relationship between these variables, with each playing a distinct role in shaping economic growth, environmental sustainability, and international trade.

**Table 5.** Granger causality test.

| Null Hypothesis | F-Statistic | *p* Value | Result | Hypothesis Tested |
|---|---|---|---|---|
| $FDI_t \longmapsto CO_{2,t}$ | 0.8911 | 0.3611 | $H_0$ not rejected | PHE |
| $GDP_t \longmapsto CO_{2,t}$ | 3.6668 | 0.0761 | $H_0$ rejected | EKC |
| $TO_t \longmapsto CO_{2,t}$ | 0.7897 | 0.3891 | $H_0$ not rejected | - |
| $GDP_t \longmapsto FDI_t$ | 2.5757 | 0.1308 | $H_0$ not rejected | - |
| $TO_t \longmapsto FDI_t$ | 0.1192 | 0.7350 | $H_0$ not rejected | - |
| $CO_{2,t} \longmapsto FDI_t$ | 3.1375 | 0.0982 | $H_0$ rejected | PHH |

Source: Author's calculation.

## 5. Conclusions

The purpose of this study was to examine the relationship between $CO_2$ emissions, FDI, economic growth and trade openess. Specifically, we were interested in confirming the presence of either the PHH or PHE. Even more so, our attention was also on confirming if the EKC was observable in Romania. The PHH suggests that countries with weaker environmental regulations and lower environmental quality are more attractive to FDI due to lower operating costs for MNEs. On the other hand, the PHE says that countries with high levels of environmental protection and quality can generate positive spillover effects for FDI by attracting MNEs seeking to enhance their reputation and brand image. By analyzing the relationships between $CO_2$ emissions, FDI, economic growth and trade openess, our study aimed to provide empirical evidence for either of these two hypotheses and shed light on the role that environmental degredation plays in attracting foreign investments.

Our research findings suggests that a linear relationship between $CO_2$ emissions, economic growth, FDI inflows and trade openess is present. FDI does seem to be a part of $CO_2$ emissions and vice versa. Groups from these variables have been observed to move together in some cases, especially $FDI_{prel}/FDI_{cons}$ and $Oil/Cement$. Manufacturing ($FDI_{prel}$) processes often involve the use of large amounts of energy, which is typically derived from fossil fuels such as $Oil$. Fossil fuels are composed primarily of carbon and hydrogen, and when burned, they release $CO_2$ into the atmosphere. Therefore, the use of $Oil$ as an energy source in manufacturing processes explains the correlation observed.

Similarly, the production of $Cement$ used in construction ($FDI_{cons}$) requires a significant amount of energy, which is primarily derived from burning fossil fuels such as $Coal$ or $Gas$. $Cement$ is made by heating limestone, clay, and other materials to extremely high temperatures in a kiln. This process, known as calcination, produces $CO_2$ as a by-product. Additionally, fossil fuels, such as coal and natural gas, are used to heat the kiln and provide the energy necessary for the production of cement. This combustion of fossil fuels also produces $CO_2$ emissions. Overall, the correlation between manufacturing and carbon emissions from oil, and construction and carbon emissions from cement, highlights the need to shift toward more sustainable energy sources and manufacturing processes to reduce the carbon footprint of these industries.

We know that PHH suggests that firms may relocate their production to countries with weaker environmental regulations in order to reduce costs and avoid stricter regulations in their home country. Our research found that there is a relationship between carbon dioxide ($CO_2$) emissions and $FDI$ in Romania, as indicated by the Granger causality test. This finding is significant because it suggests that the PHH hypothesis may hold true for Romania, and that foreign firms may be attracted to Romania due to its relatively weak environmental regulations.

Additionally, our research found that GDP does Granger-cause $CO_2$ emissions in Romania, as indicated by the EKC (Environmental Kuznets Curve) relationship. The EKC is a hypothesis that suggests that environmental degradation first increases as economic growth occurs, but it eventually decreases as countries become wealthier and shift their focus toward environmental protection. In this case, our research found that GDP growth in Romania leads to an increase in $CO_2$ emissions, but that there may be a point in the future where this trend begins to reverse and $CO_2$ emissions decrease as Romania becomes more environmentally conscious.

Overall, these findings suggest that Romania may be vulnerable to the PHH hypothesis and that it needs to prioritize environmental regulations in order to prevent further increases in carbon emissions. The findings also highlight the importance of considering the relationship between economic growth and environmental degradation and the need to transition toward more sustainable economic practices in order to reduce carbon emissions.

**Author Contributions:** Writing—original draft, A.C. (Alexandru Chiriluș); Writing—review & editing, A.C. (Adrian Costea); Supervision, A.C. (Adrian Costea). All authors have read and agreed to the published version of the manuscript.

**Funding:** This research received no external funding.

**Data Availability Statement:** The data used in the study was public data.

**Conflicts of Interest:** The authors declare no conflict of interest.

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
