# Peer review of "The Effect of FDI on Environmental Degradation in Romania: Testing the Pollution Haven Hypothesis"

_sustainability, doi:10.3390/su151310733_

Round 1
Reviewer 1 Report
Please see attached file.

English is fine.
Reviewer 2 Report
While the paper is well-written in English and includes a literature review, it lacks clarity, particularly in the econometric analysis section. Firstly, the rationale for not employing a panel approach across all industries remains unclear. Secondly, the presentation of tables diverges from conventional norms in statistical analysis, which may be confusing for readers. Thirdly, you seem to be testing a linear relationship, while the focus of your research, as implied, is on exploring a non-linear relationship over time. I recommend thoroughly revising the results section and contemplating a more appropriate approach to address the research problem. This revision should include more explicit justification for your methodological choices and a clearer presentation of your findings. Here are some comments to guide you in the right direction:
Lines 60-75 and 95-105: Several of the arguments presented here appear to be repetitive, having been discussed earlier in the text. It's advisable to revise the introduction to eliminate the redundancy and promote a more concise and effective discussion.
Line 267: To substantiate this point, consider including references from more recent literature.
Line 295: A possessive apostrophe is needed. It should read "Rezza's".
Lines 324-329: The acronyms for PHH, PHE, and FDI have been introduced already. Consider this when revising for clarity and brevity.
Lines 328-329: The claim made here lacks evidence from the literature review. It's advisable to revise this section and include supporting references.
Lines 330-332: Similar to the above, this content lacks appropriate literature support and may not fit the context of the Literature Review section. Consider revising accordingly.
Figure 1: It would be beneficial to use a vector graphic to improve the image quality. Furthermore, the placement of the figure needs adjustment. According to referencing practices, it should be placed below the first reference on page 5.
Section 3.1: Upon initial reading of the paper, the nature of the data being analyzed is unclear. It would be helpful to include a comprehensive table outlining all relevant variables, groups, and types, which would enhance the reader's understanding.
Section 3.3 Wavelet Analysis: The motivation and benefits of employing wavelet analysis in this study are not sufficiently clear. Please elucidate why this method was chosen and what advantages it offers for your specific analysis. Given that this technique might not be familiar to the typical reader of this journal, it would be beneficial to provide a more detailed explanation of the method as well.
Section 3.4 Linear Model: A simple linear regression is not typically the best tool for measuring correlation. While a linear regression can indicate the direction and strength of a linear relationship between two variables, it doesn't directly measure correlation. Correlation coefficients, such as Pearson's r, Spearman's rho, or Kendall's tau, are often better choices for quantifying the degree to which two variables are related. These correlation tests measure the strength and direction of a relationship between two variables and are not dependent on a specific variable being an independent or dependent variable.
Section 3.5: While the traditional Granger causality test does indeed assume a linear relationship between variables, it might not be the optimal choice in your case, particularly because your interest lies in examining non-linear relationships such as those suggested by the EKC and PHE. This test checks whether past values of one variable can help predict the current value of another. However, it assumes linearity, and therefore may not adequately capture non-linear relationships or interactions between variables. Using the traditional Granger causality test to analyze non-linear relationships could potentially yield misleading results. I recommend considering alternative methods better suited for investigating non-linear relationships.
Table 1: Although it might seem like a minor detail, it would be beneficial for the reader if p-values for the tests were provided to clarify the statistical significance of your findings.
Section 4.4: This section warrants further elaboration. It would be beneficial to provide more insights into the implications of these results. What can we learn from these findings? Please consider expanding on the interpretation and significance of the results to enhance understanding.
Tables 3 and 4: It is customary to consolidate these results into a single table to facilitate comprehension. The current division may cause unnecessary confusion for readers. Please consider revising these tables, including the correlation analysis in one unified table, and utilizing *, **, *** stars to denote levels of significance.
Table 4: The presentation of the regression output in its current form is unconventional. It would be advisable to revise it to match the customary style. This would include providing p-values, t-statistics, etc., for all coefficients, and the R-squared value, among other statistical measures, to ensure a comprehensive understanding of the analysis.
Lines 462-471: Although the interpretations are technically correct, they do oversimplify the interpretation of p-values. It's important to remember that a p-value does not measure the probability that the observed relationship was due to random chance, but instead measures the probability of obtaining an effect at least as extreme as the one in your sample data, assuming the null hypothesis of no effect is true. Additionally, the interpretation of the coefficients is entirely missing. The readers are left in the dark about the significance of the coefficient values. It's important to include an explanation of what these values represent in the context of your analysis, helping to convey the direct impact each variable has on the outcome.
The paper demonstrates proficient English writing skills. However, I would recommend conducting a thorough proofreading to address repetitive arguments, particularly in the introduction section.
Reviewer 3 Report
Thank you for providing an opportunity to review the draft of this manuscript on “The effect of FDI on environmental degradation in Romania: testing the pollution haven hypothesis.” The paper empirically tested the Pollution Haven Hypothesis and found that economic growth would attract FDI, contributing to environmental degradation. After thoroughly reviewing the paper, I found this paper is well-written and will significantly contribute to the policy implications.
I have the following minor observations in this paper.
· Page 7: lines 322-323 Authors summarize the key literature you reviewed. It seems confusing to a reader. I suggest the authors change it to “summary of key literature review.”
· Page 8: The manuscript is already too lengthy; I would urge authors succinctly explain the Environmental Kuznets Curve (EKC). I have seen that some diagram explanation needs to be included here. How would you link this diagram with PHE and PHH theories and your main objective of testing FDI with environmental degradation?
· If a large amount of FDI is directed toward the service industry rather than the more polluting manufacturing industry, that would positively impact the environment, such as less pollution or Co2 emission in the background. This inverted U curve mentioned that once a country achieves a certain GDP per capita growth level, it will go for clean energy, and environmental pollution will tend to decline. The nature of investment from abroad also determines the environmental degradation of the hosting country. I suggest that authors consider this point while running or analyzing the data.
· Please also insert the limitation of this study.
· What is the existing gap in the existing literature? also, please highlight the novelty of the research.
Round 2
Reviewer 2 Report
Thank you for revising your manuscript. Most of my points have been addressed satisfactorily. However, there appears to be a typo/ error in Table 4 regarding the p-value 0.7014. According to the corresponding text, it should read 0.0701. Additionally, there seems to be an issue with the ordering in the lower part of the table; currently, both elements (obs, R^2 etc) are listed under CO2, which appears to be incorrect.
Author Response
The typos in Table 4 have been corrected.
Thank you for your support.